# Automated Deep Learning-Based Classification of Wilms Tumor Histopathology

**DOI:** 10.3390/cancers15092656

**Published:** 2023-05-08

**Authors:** Ananda van der Kamp, Thomas de Bel, Ludo van Alst, Jikke Rutgers, Marry M. van den Heuvel-Eibrink, Annelies M. C. Mavinkurve-Groothuis, Jeroen van der Laak, Ronald R. de Krijger

**Affiliations:** 1Princess Máxima Center for Pediatric Oncology, Heidelberglaan 24, 3584 CS Utrecht, The Netherlands; 2Department of Pathology, Radboud University Medical Center, Geert Grooteplein 1, 6500 HB Nijmegen, The Netherlands; 3Center for Medical Image Science and Visualization, Linköping University, 581 83 Linköping, Sweden; 4Department of Pathology, University Medical Center Utrecht, Heidelberglaan 100, 3584 CX Utrecht, The Netherlands

**Keywords:** artificial intelligence, Wilms tumor, pediatric pathology, deep-learning, tumor segmentation

## Abstract

**Simple Summary:**

Wilms tumor (WT) is the most frequent pediatric tumor in children and shows highly variable histology, leading to variation in classification. Artificial intelligence-based automatic recognition holds the promise that this may be done in a more consistent way than human observers can. We have therefore studied digital microscopic slides, stained with standard hematoxylin and eosin, of 72 WT patients and used a deep learning (DL) system for the recognition of 15 different normal and tumor components. We show that such DL system can do this task with high accuracy, as exemplified by a Dice score of 0.85 for the 15 components. This approach may allow future automated WT classification.

**Abstract:**

(1) Background: Histopathological assessment of Wilms tumors (WT) is crucial for risk group classification to guide postoperative stratification in chemotherapy pre-treated WT cases. However, due to the heterogeneous nature of the tumor, significant interobserver variation between pathologists in WT diagnosis has been observed, potentially leading to misclassification and suboptimal treatment. We investigated whether artificial intelligence (AI) can contribute to accurate and reproducible histopathological assessment of WT through recognition of individual histopathological tumor components. (2) Methods: We assessed the performance of a deep learning-based AI system in quantifying WT components in hematoxylin and eosin-stained slides by calculating the Sørensen–Dice coefficient for fifteen predefined renal tissue components, including six tumor-related components. We trained the AI system using multiclass annotations from 72 whole-slide images of patients diagnosed with WT. (3) Results: The overall Dice coefficient for all fifteen tissue components was 0.85 and for the six tumor-related components was 0.79. Tumor segmentation worked best to reliably identify necrosis (Dice coefficient 0.98) and blastema (Dice coefficient 0.82). (4) Conclusions: Accurate histopathological classification of WT may be feasible using a digital pathology-based AI system in a national cohort of WT patients.

## 1. Introduction

Wilms tumor (WT), also called nephroblastoma, is the most common type (80–90%) of pediatric renal tumor [1]. As a result of improved stratification, randomized trials, improved treatment options, and enhanced supportive care, the 5-year survival rate has increased from less than 30% in the 1930s to more than 90% at present. However, survival rates have dropped to less than 30% in case of recurrence. Wilms tumors are heterogeneous, consisting of blastemal, stromal and epithelial components. Relapses are more common in certain WT subtypes, such as those with large blastemal components after preoperative chemotherapy and those with diffuse anaplasia, also identified as high-risk groups based on several large-scale epidemiological studies [2,3]. Furthermore, toxic effects later in life are reported in 25% of WT patients and particularly occur in those who received high-dose chemotherapy in combination with radiotherapy [4,5]. Hence, to prevent under- and overtreating patients, histopathological classification has become crucial for proper treatment stratification and subsequent treatment choice. In the International Society of Pediatric Oncology (SIOP) 2016 UMBRELLA protocol, this classification is response-based, following pre-operative chemotherapy, in contrast to the histology-based classification of the USA National Wilms Tumor Study Group (NWTSG)/Children’s Oncology Group (COG) which is applied after direct surgery [6,7]. Nevertheless, it has been reported that quantifying WT components using conventional histopathological assessment has low reproducibility, and that this is subject to interobserver variability [8]. Artificial intelligence (AI) could possibly contribute to a more accurate and reproducible quantification and thereby improve risk stratification. Machine learning (ML) is an AI technique that uses mathematical algorithms to learn from experience without being extensively supervised by humans. Especially, deep learning (DL), a subtype of ML that uses multi-layered filtering steps called neural networks (NNs), shows great promise in healthcare applications [9,10,11]. For image recognition, the most commonly used model of NN is the Convolutional Neural Networks (CNNs), which have the ability to extract meaningful image features from pixels and on the basis thereof derive a specific category [12]. In contrast to other medical fields, such as dermatology [13], radiology [14] cardiology [15], and urology [16], where AI has been widely implemented, the use of AI in pathology is still relatively new, and it mainly focuses on common adult cancer types. Especially in radiology, where large datasets have been available for a long time, AI has shown great potential. For instance, Yadav et al. [17] showed that in radiology AI could be beneficiary for the classification of brain tumors; Hameed et al. [16] displayed the wide range of using AI in the early diagnosis, treatment, and classification of urological diseases; whereas Khan et al. [18] used AI in neuroimaging for the early detection of Alzheimer’s disease. The introduction of digitization of glass slides, called whole-slide imaging (WSI) paved the way for digital pathology [19]. WSI provides the ability to annotate large datasets and to analyze multiple images using DL. For example, a recent study on renal pathology showed that the use of a DL-based system can assist and enhance the quality of classification [20]. Hermsen et al. [21] developed a DL-based system that sustained a high segmentation performance in kidney components. However, to provide a dataset that is large enough, time-consuming manual annotations are needed. In our previous work, we have extensively annotated WTs to provide a sufficiently large enough dataset [22].

In the current study, the performance of a DL-based system in recognizing normal and tissue components on digital histological images in a series of Wilms tumors is assessed. Although AI has been previously investigated in assessing urological diseases and renal pathology, this will be the first study conducted on WT.

## 2. Materials and Methods

### 2.1. Study Design and Population

This study was conducted on 105 patients from a national cohort of WT in the Netherlands [23]. All children that were diagnosed between 2014 and 2019, younger than 18 years, treated according to the SIOP 2001 or UMBRELLA protocol and with informed consent were included. Approval by the Medical Ethics Committee (MEC 202.134/2001/122, MEC-2018-026, and MEC-2006-348, and Netherlands Trial Register NL7744 with ethics committee approval number MEC-2016-739) was obtained. Representative hematoxylin and eosin (HE)-stained glass slides were retrieved from the pathology archives and digitized. Patients’ demographics were retrieved from the patient registry of the Princess Máxima Center, and histopathology findings were based on conventional pathology according to SIOP classification [24].

### 2.2. Image Data Sets

Whole-slide images (WSIs) of WT in HE-stained tissue sections were generated using a digital slide scanner (Pannoramic P1000; 3DHISTECH Ltd., Budapest, Hungary) with a 20× objective and adapter (additional magnification 1.6×) at a resolution of 0.24 μm/pixel. The dataset contained 1181 glass slides. Slides of metastases, biopsies, multiple tumors, and primary nephrectomy specimens without preceding chemotherapy were excluded from this dataset. Eleven slides were excluded due to poor slide quality.

### 2.3. CNN Development and Design

For the purpose of the current study, twenty WSI’s were randomly selected. Using the automated slide analysis platform software (ASAP), a total of 4995 manual annotations with 19 predefined classes were developed, which included vital tumor components, chemotherapy-induced changes and normal renal tissue, as depicted in Table 1 [22]. Because of insufficient data, we decided to exclude four components (background, anaplasia, adrenal medulla and adrenal cortex). To facilitate accurate distinction between the fifteen remaining classes, two additional overarching classes “tumor” and “non-tumor” were included. The 4995 annotations were divided into 70% training (to train the algorithm), 15% validation (to fine-tune the algorithm) and 15% testing annotation (to evaluate the performance of the algorithm) sets. Particular attention was paid to the fact that annotations from cases used in one set were excluded from the other sets. From the annotations, a large dataset of patches were extracted to train the model. Two CNN architectures (U-net and DenseNet) commonly used in segmentation tasks were trained, and their performance was compared to find the most optimal CNN for this task. The neural networks were implemented according to Ronneberger et al. and Huang et al., for the U-net and DenseNet, respectively [25,26]. We trained both neural networks for 200 epochs. Each epoch consisted of 600 iterations with a batch size of 16 for the DenseNet and 4 for the U-net. During training, patches measuring 412 × 412 for the U-net and 128 × 128 for the DenseNet were randomly sampled from the annotated regions at a resolution of 0.5 μm/pixel. The network was optimized using Adam with a categorical cross-entropy loss [27]. The initial learning rate was set at 0.0005 and was reduced by a factor of two after a loss plateau of 5 epochs. In addition to standard rotation/flipping, we applied Gaussian blurring and color augmentations to the patches during training to improve the networks generalization to variations not seen in the training set. The final layer of both neural networks was a softmax layer. To get an output of each network, we picked the class with the highest probability after the softmax layer for each pixel in the input image. Outputs for the WSIs were generated for both networks by slicing the image into smaller patches and putting each patch through the network, after which the outputs were stitched together to get an output for each pixel in the WSI. An overview of the network architectures can be seen in Table 2. The models were implemented in python (3.6) using Tensorflow (1.15).

### 2.4. Histopathological Classification

The conventional histopathological assessment of WT in HE-stained tissue was performed by an experienced pathologist (member of the international SIOP-RTSG WT panel) and a second review was conducted by a local, but experienced pediatric pathologist. To classify each WT according to the SIOP-RTSG classification, the pathologist determined the percentage of blastema, epithelium, stroma, and of therapy-induced changes [7]. Examples of the histopathological classification are shown in Figure 1.

### 2.5. Statistical Analysis

SPSS was used for the descriptive statistics. The performance of the algorithm was evaluated using the Sørensen–Dice coefficient (F1 score), defined as the ratio of correctly labeled pixels and the total number of pixels (recall score). The highest score was 1.0 (all pixels correctly labeled) and the lowest score was 0 (no pixels correctly labeled). The performance was visualized with a confusion matrix, which showed the correct and incorrect predictions. The values on the diagonal represent the fraction of correctly predicted samples for each type of tissue.

## 3. Results

### 3.1. Study Population

Out of 105 WT cases, a total of 72 patients met the inclusion criteria; patients with insufficient slides, metastatic cases, biopsies, primary nephrectomy specimens and patients with multiple tumors were excluded. These 72 patients were classified and treated according to the SIOP 2001 or SIOP-RTSG 2016 UMBRELLA protocol [7]. Table 3 and Table 4 show an overview of the patient characteristics and tumor types. The median age at diagnosis was 51 months (SD 41) and 58% of patients (*n* = 42) were female. WTs were histopathologically classified as low risk in 3% (*n* = 2), intermediate risk in 90% (*n* = 65) and high-risk WT in 7% (*n* = 5). All of the 72 patients presented with localized stages; 35% (*n* = 25) stage I WT, 29% (*n* = 21) with stage II, and 36% (*n* = 26) with stage III. The majority was diagnosed with regressive type, 32% (*n* = 23) or mixed type, 34% (*n* = 25).

### 3.2. Algorithm Output

For the final result, the output of the DenseNet and U-net was combined, with individual overall Dice scores of 0.7721 and 0.7083, respectively. The networks were combined by taking the average output of the individual neural networks for each pixel in the input. This yields a slight improvement over the performance of each individual network, with a Dice score of 0.7767. Figure 2 gives an example of the outputs for DenseNet and U-net, illustrating the differences between the two networks.

### 3.3. Algorithm Performance

The classification results are presented in Figure 3. The confusion matrix shows the classification accuracy for the different classes, as a percentage of correctly labeled pixels. The CNN classified pixels were labeled with the histopathological tissue class of blastema (recall 0.96), stroma (0.59), epithelium (0.38), necrosis (0.99), bleeding (0.92) and regression (0.77). The overall recall among the fifteen tissue components ranged from 0.38 for epithelium to 1.0 for glomeruli. Incorrect classification of stromal regions occurred mostly in the category of connective tissue (0.23), whereas connective tissue was mostly incorrectly classified as regression (0.32). Incorrect classification of epithelium mostly occurred in the category of blastema (0.29).

An overview of the CNN performance is depicted in Table 5. The CNN yielded Dice scores for the histopathological components of 0.82 (blastema), 0.67 (stroma), 0.48 (epithelium), 0.98 (necrosis), 0.69 (regression) and 0.36 (bleeding). The Dice score for tissue components ranged from 0.37 (bleeding)–1.00 (glomeruli). The overall Dice score of distinguishing the fifteen tissue components is 85% (0.85), and of distinguishing vital tumor components is 70%. Bleeding yielded a high number of false positives (precision of 0.23), whereas epithelium revealed a high number of false negatives (recall of 0.38).

## 4. Discussion

In this study, we determined whether a DL-based algorithm can be applied to accurately classify tissue elements of WT compared to the conventional assessment by a pathologist. The overall Dice score was 85% for the DL-based system for distinguishing fifteen tissue components. For vital tumor components this was 70% and for chemotherapy-induced changes the Dice score was 84%. This indicates that the overall AI performance is considered sufficiently high to be used for diagnostic applications. Nevertheless, AI-based recognition of vital tumor components will benefit from further optimization.

Our system had a high accuracy in quantifying blastema (0.82). However, due to the heterogeneous nature of Wilms tumors reflected in the HE slides, classification of specific tissue elements was not always optimal. For example, due to preoperative chemotherapy (according to the SIOP-RTSG policy), it is known that the distinction between viable tumor stroma and paucicellular tissue following chemotherapy, that is classified as regression, can be very hard to recognize and thus to annotate, even for experienced pathologists [28,29]. In addition, normal connective tissue outside the tumor may look identical to regressive tissue within the tumor. These are the challenges with a DL-based system that derives its categorization from recognition of individual pixels (i.e., being blind for the larger tissue context), and has to learn from pathologists. Our results show that in cases where stroma was misclassified, it was mostly incorrectly classified as connective tissue (0.23), regression mostly as connective tissue (0.10) or stroma (0.08), and connective tissue as regression (0.32) or stroma (0.07). Histopathological assessment plays an important role in current risk group stratification of patients and is important for the future reliable application of our findings. As an example, immature epithelium consisting of tightly packed tubular structures with considerable nuclear pleomorphism may strongly resemble blastema and may thus be hard to distinguish from it [28]. However, epithelial WT is currently classified as intermediate risk histology in contrast to blastema type which is classified as high risk, and is related to a higher recurrence risk. The importance is exemplified in our system in which there was a suboptimal recognition of epithelium (Dice score of only 48%) and when epithelium was misclassified, it was mostly incorrectly classified as blastema (0.29) [28]. This could potentially lead to misclassification as a higher risk group and therefore result in treating WT patients too intensively. However, before such a statement can be made, formal comparison of risk group classification on the basis of this AI system and by pathologists on a sufficiently large set of cases needs to be pursued.

The current study is different from previous DL approaches regarding urological diseases/renal pathology. Hara et al. developed a U-net to evaluate tubulointerstitial compartments to classify normal and abnormal renal tubes [20]. Hameed et al. showed the wide range of DL-based radiology applications used in urological diseases, ranging from robotic assisted surgery in adults to predicting the benefits of performing a voiding cystourethrogram in children suspected of having vesicoureteral reflux [16]. Hermsen et al. [21] developed a U-net for multiclass segmentation of kidney tissue with a Dice score ranging from 0.8–0.84. This is the first study conducted on the classification of histopathological components of WT.

There are several limitations to this study. Firstly, this is the first study that investigates automated quantification of WT components, and it is conducted on a limited series of patients. Therefore, validation in a larger cohort is required. Secondly, anaplasia and nephrogenic rests were not included in the DL-based system because of the low number of patients and the rare occurrence of these features. Determining the presence of diffuse anaplastic WT is often challenging for pathologists. However, diffuse anaplasia, which is based on the simultaneous presence of three criteria, enlarged nuclei, hyperchromasia of nuclei and atypical mitoses, represents an adverse prognostic group and is therefore important to be recognized [28]. Criteria for anaplasia are challenging and teaching this feature to a DL-based system requires an alternative approach based on point analysis rather than the field analysis, which was employed here for the 15 tissue classes that were studied. Thus, analysis of anaplasia was beyond the scope of this study but should be considered in future studies.

The last limitation of the design of this study is that we only investigated tissue components of WT without reference to other anatomical structures, such as the renal sinus and resection margins. Additionally, lymph nodes in resection specimens were outside the scope of this study. However, for current risk treatment stratification, both tumor staging and risk group assignment on the basis of histopathological annotations are needed. Therefore, before AI-based patient risk classification can be pursued and translated towards treatment choices, further research including validation in a larger cohort including relevant subtypes, such as diffuse anaplasia and nephroblastomatosis is required.

## 5. Conclusions

This first AI-translational study on WT histological assessment and automated recognition of normal and tumor histological components showed an overall high performance of the DL-based system with a Dice score of 85%, suggesting its applicability in clinical practice for quantification of these rare tumors. Further research maybe conducted to use a DL-based system in risk group classification in routine pathology. Further research including the detection of anaplasia, renal sinus characteristics, lymph node histology and other staging criteria, as well as a more accurate prediction of blastemal, epithelial cells and nephroblastoma is required. Studies in parallel to a conventional risk classification that include large international cohort studies (such as in SIOP-RTSG) with appropriate treatment and outcome data are necessary to validate such an approach before it is feasible to use in clinical practice.

## Figures and Tables

**Figure 1 cancers-15-02656-f001:**
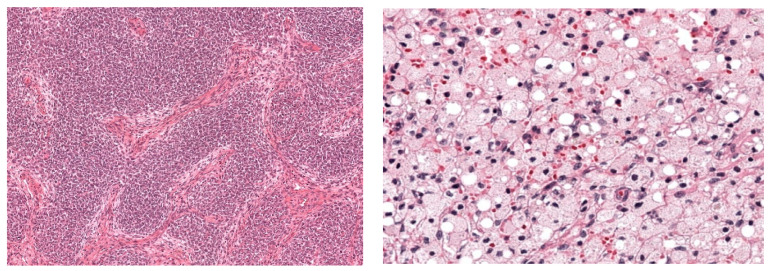
Hematoxyin and eosin-stained sections showing histopathological components for WT. The left image shows blastemal, one of the three recognized tumor components; the right image shows regression due to chemotherapy pretreatment.

**Figure 2 cancers-15-02656-f002:**
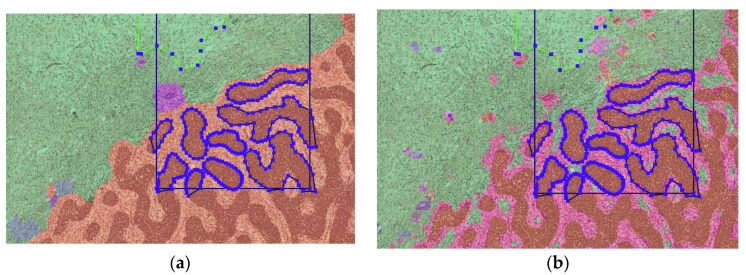
(**a**) Prediction of DenseNet (**b**) Prediction of U-net in the same region, with some ground truth annotations also depicted. In the image we see WT stroma (pink) interspersed WT blastema (orange). This is correctly classified by the U-net (**b**), but not by the DenseNet (**a**), where no pixels were classified as WT stroma (pink). However, the U-net is slightly more noisy in the normal stromal tissue (green) than the DenseNet.

**Figure 3 cancers-15-02656-f003:**
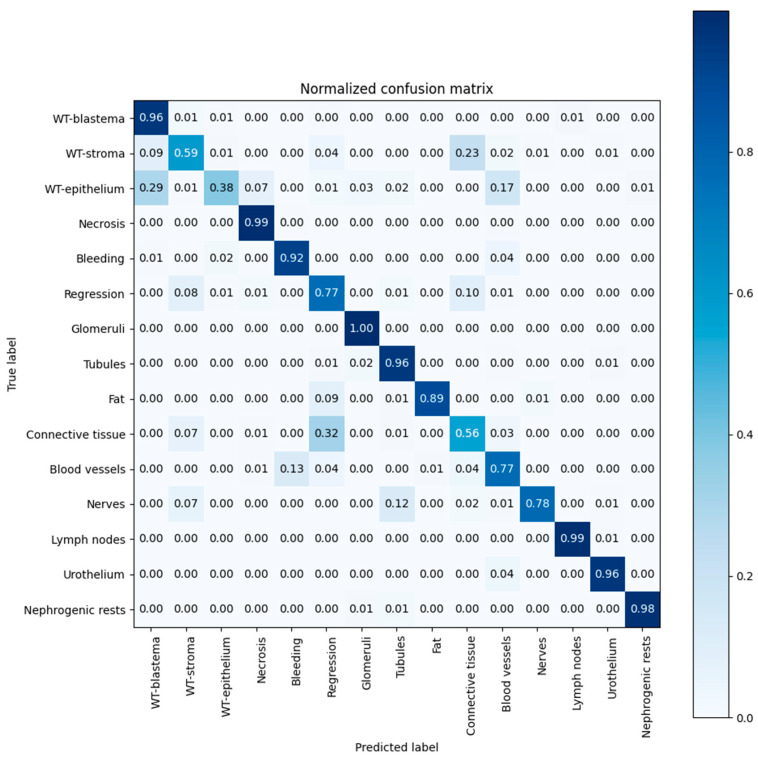
Confusion matrix for the ensemble of DenseNet and U-net. This shows which tissue types are being confused with each other; the values on the diagonal represent the fraction of correctly classified pixels for each type of tissue.

**Table 1 cancers-15-02656-t001:** Annotated tissue components.

Vital Tumor Components	Chemotherapy-Induced Changes	Normal Renal Tissue
Blastema Stroma Epithelium	Necrosis Bleeding Regression	Glomeruli Tubules
**Extra Renal Tissue**	**Adrenal Gland**	**Others**
Fat Mesenchyme Vessels Nerves Lymph nodes	Adrenal cortex Adrenal medulla	Urothelium Anaplasia Nephrogenic rest Background

**Table 2 cancers-15-02656-t002:** Network architectures.

	U-net Training Defaults	DenseNet Training Defaults
**Patch shape**	(412, 412)	(128, 128)
**Sampling spacing**	0.5 μm/pixel	0.5 μm/pixel
**Loss function**	categorical cross-entropy	categorical cross-entropy
**Optimization method**	Adam [27]	Adam [27]
**Epochs trained**	200	200
**Batch size**	4	16
**Initial learning rate**	0.0005	0.0005
**Learning rate decay**	0.5 after plateau of 5 epochs	0.5 after plateau of 5 epochs
**Augmentations used**	rotation, flipping, Gaussian noise and color	rotation, flipping, Gaussian noise and color
**Final layer**	Softmax	Softmax

**Table 3 cancers-15-02656-t003:** Patient characteristics.

Patient Demographics (*n* = 72)	*n* (%)
Age in months at time of diagnosis, Mean (SD)	51.4 (±41.3)
Female gender	42 (58.3)
Left-sided WT localization	41 (56.9)
Lymph node metastases	11 (15.3)
**Histology * (*n* = 72)**	*n* (%)
Low risk	2 (2.8)
Intermediate risk	65 (90.2)
High risk	5 (6.9)
**Tumor histology * (*n* = 72)**	*n* (%)
Completely necrotic^1^	2 (2.8)
Regressive ^2^	23 (31.9)
Epithelial ^2^	5 (6.9)
Stromal ^2^	12 (16.7)
Mixed ^2^	25 (33.8)
Blastemal ^3^	5 (6.9)
**SIOP overall stage (*n* = 72)**	*n* (%)
I	25 (34.7)
II	21 (29.2)
III	26 (36.1)

* Tumor classification according to SIOP-2016. Low-risk 1; Intermediate-risk 2; High-risk 3.

**Table 4 cancers-15-02656-t004:** Histopathological characteristics.

Tumor Characteristics (*n* = 72)	*n* (%)
Chemotherapy-induced changes	
<66%	47 (65.3)
>66%	23 (31.9)
100%	2 (2.8)
Blastema	
<66%	62 (86.1)
>66%	8 (11.1)
Missing	2 (2.8)
Epithelium	
<66%	64 (88.9)
>66%	7 (9.7)
Missing	1 (1.4)
Stroma	
<66%	57 (79.2)
>66%	13 (18.1)
Missing	2 (2.8)

**Table 5 cancers-15-02656-t005:** Accuracy of algorithm in terms of dice coefficient.

Tissue Element	Precision	Recall	Dice Coef.
WT-blastema	0.71	0.96	0.82
WT-stroma	0.77	0.59	0.67
WT-epithelium	0.65	0.38	0.48
Necrosis	0.98	0.99	0.98
Bleeding	0.23	0.92	0.37
Regression	0.62	0.77	0.69
Glomeruli	0.69	1.00	0.82
Tubules	0.98	0.96	0.97
Fat	1.00	0.89	0.94
Mesenchyme	0.57	0.67	0.62
Vessels	0.85	0.77	0.81
Nerves	0.85	0.77	0.81
Lymph nodes	0.99	0.99	0.99
Urothelium	0.46	0.96	0.62
Nephrogenic rests	0.82	0.98	0.89
Chemotherapy-induced changes	0.79	0.90	0.84
Vital tumor components	0.74	0.66	0.70
Overall score	0.85	0.85	0.85

## Data Availability

Reader of this manuscript are welcome to contact the authors of this manuscript for further details off the data.

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
