# Peer review of "Automated Deep Learning-Based Classification of Wilms Tumor Histopathology"

_cancers, 2023, doi:10.3390/cancers15092656_

Round 1
Reviewer 1 Report
1. What do you mean by recognition in the title? Is it prediction or classification?
2. Keywords - Minimum 5 keywords should be presented.
3. The introduction section can be further strengthened with reference to the following articles and various applications of Deep learning in the field of healthcare. Several researchers have addressed various methods to do the same. Kindly highlight the novelty addressed in the present study.
https://doi.org/10.3390/jimaging9020033
https://umcg.studenttheses.ub.rug.nl/2908/
https://doi.org/10.3390/jcm10091864
https://doi.org/10.3390/jimaging9010010
https://doi.org/10.1371/journal.pone.0271161
https://doi.org/10.1172/jci.insight.144779
3. The authors should include a comparison table of previous studies' results and present study outcomes in discussion section.
4. Kindly check Table 1. The numbering looks erroneous. The numbers can be removed
5. Methodology or approach of CNN based algorithm used should be further explained. Including the platform and the flowchart would clarify how feature extraction was carried out and what makes it automated.
6. The writing should be improved. Kindly use statements like “the present study”, rather than using sentences such as “we did”, “we decided”, etc. Kindly revise it throughout the manuscript.
7. The couple of images representing Wilms Tumor histopathology obtained from a digital slide scanner can be included with the markings of region of interest the deep learning neural network should identify.
Author Response
Reviewer 1
- What do you mean by recognition in the title? Is it prediction or classification?
Thank you for this remark. We indeed mean classification. We have adapted the title accordingly.
- Keywords - Minimum 5 keywords should be presented.
We have added two key words: deep learning; tumor segmentation
- The introduction section can be further strengthened with reference to the following articles and various applications of Deep learning in the field of healthcare. Several researchers have addressed various methods to do the same. Kindly highlight the novelty addressed in the present study.
We have expanded the introduction by introducing a paragraph related to the references that were kindly provided by this reviewer (see introduction lines 62-66 and 69-72). We note that one of the references in fact is the thesis work of the first author. In addition, we have highlighted the novelty of the present study at the end of the introduction in lines 78-79.
- The authors should include a comparison table of previous studies' results and present study outcomes in discussion section.
We thank the reviewer for this remark. As this is not a review article, we have chosen not to make a table of the results of previous studies, but instead have devoted a paragraph in the introduction (see lines 62-66 and 69-72) and in the discussion (see lines 295-303).
- Kindly check Table 1. The numbering looks erroneous. The numbers can be removed
We have removed the numbers.
- Methodology or approach of CNN based algorithm used should be further explained. Including the platform and the flowchart would clarify how feature extraction was carried out and what makes it automated.
We have added the platform in which the algorithms were implemented. We also added training details for both network architectures (see lines 117-132 and table 2).
- The writing should be improved. Kindly use statements like “the present study”, rather than using sentences such as “we did”, “we decided”, etc. Kindly revise it throughout the manuscript.
We have revised the manuscript with particular attention to the writing by an experienced scientist with degree in English language.
- The couple of images representing Wilms Tumor histopathology obtained from a digital slide scanner can be included with the markings of region of interest the deep learning neural network should identify.
We have added two images of annotated histological images indicating various histopathological components, to be recognized by the CNN (please see results, new Figure 1 on page 5).

Reviewer 2 Report
A well organized paper on an interesting topic. But I think few updates are required to improve this paper:
(1) In Results Section, subsection Algorithm is too short and unclear. For readers, authors must mention pseudo code or algorithm here.
(2) Why Figure 1 in should not come in the Algorithm (subsection 3.2). It may be added in a new subsection "Result Analysis". Also, explanation is not enough.
(3) In the References there are 24 references but only 19 are cited in the text.
(4) Any architecture diagram of the proposed work may be added.
(5) Include more references with latest papers. Include one relevant in your reference: Khan, Y. F., Kaushik, B., Chowdhary, C. L., & Srivastava, G. (2022). Ensemble Model for Diagnostic Classification of Alzheimer’s Disease Based on Brain Anatomical Magnetic Resonance Imaging. Diagnostics, 12(12), 3193.
Author Response
Reviewer 2
A well-organized paper on an interesting topic. But I think few updates are required to improve this paper:
- In Results Section, subsection Algorithm is too short and unclear. For readers, authors must mention pseudo code or algorithm here.
We have added the platform in which the algorithms were implemented. We also added training details for both network architectures (see lines 117-132 and table 2).
- Why Figure 1 in should not come in the Algorithm (subsection 3.2). It may be added in a new subsection "Result Analysis". Also, explanation is not enough.
We added some additional details to this, which has now become figure 2, as a result of inserting a new figure (see lines 211 – 212 and 214-215).
- In the References there are 24 references but only 19 are cited in the text.
We have checked this and adapted where necessary (also on the basis of additional references as suggested by both reviewers). As far as we can see, all references are now cited in the text.
- Any architecture diagram of the proposed work may be added.
We have added the platform in which the algorithms were implemented. We also added training details for both network architectures (see lines 117-132 and table 2). We added thorough training details for both of the network architectures (DenseNet and U-net).
- Include more references with latest papers. Include one relevant in your reference: Khan, Y. F., Kaushik, B., Chowdhary, C. L., & Srivastava, G. (2022). Ensemble Model for Diagnostic Classification of Alzheimer’s Disease Based on Brain Anatomical Magnetic Resonance Imaging. Diagnostics, 12(12), 3193.
We have added this reference, together with those suggested by reviewer 1 and integrated these in the introduction (see lines 62-66 and 69-72) and in the discussion (see lines 295-303).

Round 2
Reviewer 2 Report
All the review comments were addressed by the authors